# Intensity scaling of conventional brain magnetic resonance images avoiding cerebral reference regions: A systematic review

Tun Wiltgen[1,2]*, Cuici Voon[1,2], Koen Van Leemput[3,4], Benedikt Wiestler[5], Mark Mühlau[1,2]*

1 Department of Neurology, School of Medicine, Technical University of Munich, Munich, Germany, 2 TUM-Neuroimaging Center, School of Medicine, Technical University of Munich, Munich, Germany, 3 Department of Neuroscience and Biomedical Engineering, Aalto University Helsinki, Espoo, Finland, 4 Athinoula A. Martinos Center for Biomedical Imaging, Massachusetts General Hospital, Harvard Medical School, Boston, Massachusetts, United States of America, 5 Department of Neuroradiology, School of Medicine, Technical University of Munich, Munich, Germany

* tun.wiltgen@tum.de (TW); mark.muehlau@tum.de (MM)

## Abstract

### Background

Conventional brain magnetic resonance imaging (MRI) produces image intensities that have an arbitrary scale, hampering quantification. Intensity scaling aims to overcome this shortfall. As neurodegenerative and inflammatory disorders may affect all brain compartments, reference regions within the brain may be misleading. Here we summarize approaches for intensity scaling of conventional T1-weighted (w) and T2w brain MRI avoiding reference regions within the brain.

### Methods

Literature was searched in the databases of Scopus, PubMed, and Web of Science. We included only studies that avoided reference regions within the brain for intensity scaling and provided validating evidence, which we divided into four categories: 1) comparative variance reduction, 2) comparative correlation with clinical parameters, 3) relation to quantitative imaging, or 4) relation to histology.

### Results

Of the 3825 studies screened, 24 fulfilled the inclusion criteria. Three studies used scaled T1w images, 2 scaled T2w images, and 21 T1w/T2w-ratio calculation (with double counts). A robust reduction in variance was reported. Twenty studies investigated the relation of scaled intensities to different types of quantitative imaging. Statistically significant correlations with clinical or demographic data were reported in 8 studies. Four studies reporting the relation to histology gave no clear picture of the main signal driver of conventional T1w and T2w MRI sequences.

**Data Availability Statement:** All relevant data are within the manuscript and its Supporting information files.

**Funding:** This study has received funding by a research grant of the National Institutes of Health (grant 1R01NS112161-01) and by a research grant of the German Research Foundation, DFG Priority Programme 2177, Radiomics: Next Generation of Biomedical Imaging (grant 428223038). The funders had no role in study design, data collection and analysis, decision to publish, or preparation of the manuscript.

## Conclusions

T1w/T2w-ratio calculation was applied most often. Variance reduction and correlations with other measures suggest a biologically meaningful signal harmonization. However, there are open methodological questions and uncertainty on its biological underpinning. Validation evidence on other scaling methods is even sparser.

## 1 Introduction

Conventional T1-weighted (T1w) and T2-weighted (T2w) magnetic resonance imaging (MRI) have been used in clinical settings since the early days of MRI. The development of these sequences primarily aimed at high tissue contrast to aid visual inspection. In other words, conventional MRI sequences are optimized for high local contrast rather than for consistency of intensity values over the scan volume or over time. The absolute intensity values of T1w and T2w images can indeed vary largely due to several technical factors (e.g., MR hardware, amplifier gain, patient anatomy); they also have an arbitrary scale. Thus, intensity-based quantitative analysis of T1w or T2w images is not easily feasible. However, intensity variations of conventional MRI also contain biologically meaningful signals. For example, myelin maturation during brain development is perfectly paralleled by the T1w signal change in humans [1, 2], which has been confirmed histologically in pigs [3]. Therefore, intensity scaling of conventional MRI presents an attractive candidate to convert images with highly variable intensities with arbitrary scales into images with semi-quantitative intensity values, making it possible to analyze intensity variations of conventional MRI for biological questions. Given the broad use of conventional brain MRI, valid methods for intensity scaling could be easily translated into clinical practice and applied for the analyses of large-scale scientific databases. An essential prerequisite is the bias field correction, which removes local intensity inhomogeneities within MR images. Particularly, in studies investigating image intensities, local intensity inhomogeneities induced by transmit and receiver field inhomogeneities should be removed through bias field correction because observed signal variations (after intensity scaling) should arise from biological differences of tissues rather than from technical factors. Bias field correction has been established as a pre-processing step in many image processing pipelines (e.g., SPM) [4]. Although it reduces technically induced local intensity variance, it cannot harmonize arbitrary intensity scales and remove inter-scan variability.

Scaling conventional bias-corrected MRI can be achieved by histogram matching or scaling according to the overall mean image intensity [5–9]. The use of a biologically defined reference region, such as white matter or normal-appearing parts of it [10], constitutes an intuitive approach. The rationale is that a reference region, assumed to have equal biological properties across all subjects, should, in theory, lead to the same MRI signal, given that no technically induced variance exists. Hence, simple linear scaling according to the reference region should remove the technically induced intensity variance while preserving biologically meaningful variance. A reference region within the brain seems reasonable in circumscribed pathologies such as neoplasms, but it does not seem reasonable in systemic degenerative or inflammatory disorders since disease-related effects may even change the biological properties of the reference region, whose properties are no longer the same across subjects with varying disease burden. For example, in Multiple Sclerosis (MS), the entire brain can be affected, and even the normal-appearing white matter (on MRI) already shows histological changes [11], demonstrating the importance of avoiding cerebral reference regions, which can be liable candidates in these cases.

Originally, we planned to include studies using an extracerebral reference region for intensity scaling. However, the literature search revealed many studies that applied intensity scaling without reference regions but by using the T1w/T2w-ratio introduced by Glasser and van Essen [12]. We found several versions of this approach, either with or without prior scaling of the T1w and T2w images with extracerebral reference regions. Hence, we decided to extend the scope of our review and included studies using the T1w/T2w-ratio with or without prior intensity scaling if the validity of the approach was investigated. Calculation of T1w/T2w-ratio images does not require a reference region, and by dividing T1w by T2w image intensities, it is possible to obtain images with new contrasts. Most of these studies assume that the ratio of T1w and T2w images reflects myelin [12], since voxels in highly myelinated regions have high-intensity values in T1w images but low-intensity values in T2w images, but did not evaluate the method further. In addition, it has been hypothesized that T1w and T2w images are similarly affected by technical artifacts, such as the receiver bias field (B1-), and that, therefore, by calculating the ratio of T1w and T2w images, these technical artifacts are canceled out at the same time [12–14]. Yet, this assumption cannot be made for the transmit bias field (B1+), and does not seem to be true for all MRI scanners; in consequence, some authors suggested separate scaling of T1w and T2w images before calculating the ratio of the two [14].

This systematic review includes original publications on approaches for intensity scaling of conventional T1w and T2w brain MRI without using a cerebral reference region. As the application of a scaling method per se does not provide evidence of the validity of the approach, we defined inclusion criteria requiring analyses of the validity of the scaling method. To be included, studies had to provide comparative data regarding the variance reduction of intensities induced by other scaling methods including brain reference regions (e.g., histogram matching or scaling according to overall mean intensity). In many circumstances, brain reference regions can be a reasonable option for intensity scaling. This enables us to evaluate the performance of scaling methods avoiding brain reference regions. We aimed to analyze if the scaling methods of interest in this review (those avoiding cerebral reference regions) lead to similar variance reduction as established methods. Alternatively, studies had to provide the relation to other imaging techniques or the relation to histology.

## 2 Materials and methods

This review's reporting and structure rely on the Preferred Reporting Items for Systematic Reviews and Meta-Analyses 2020 (PRISMA 2020) [15]. The main review questions, search strategy, eligibility criteria, and data extraction strategy had been defined beforehand, and a detailed review protocol was registered on PROSPERO [16]. Due to different research questions, heterogeneous selection of analysis, and paradigms between studies, we could not perform a meta-analysis.

We conducted the literature search on 1 November 2022 on the following web databases: Scopus, PubMed, and Web of Science. Our search settings included combinations of the terms "magnetic resonance", "imag*", "intensit*", "normali*", "brain*", and combinations of synonyms thereof. The exact settings for each database are presented in Table 1.

In addition, we conducted a forward and backward citation search for all the included studies on 5 November 2022 using Web of Science. We specified the eligibility criteria according to the PICO framework during the pre-registration process of the review protocol.

- (P) Patient Population: Humans, no further restrictions.

- (I) Intervention (method under investigation): Methods that avoid cerebral reference regions for intensity scaling of conventional T1w and T2w brain MR images.

**Table 1. Literature search settings.**

| database | Search settings |
|---|---|
| Scopus | TITLE-ABS-KEY (((magnetic W/3 resonan*) OR t1* OR mr*) AND imag* AND intensit* AND (calibr* OR quantif* OR normali* OR standardi*) AND (brain* OR head* OR cerebr*)) |
| PubMed | (magnetic resonan*[Title/Abstract] OR t1*[Title/Abstract] OR mr*[Title/Abstract]) AND imag* [Title/Abstract] AND intensit*[Title/Abstract] AND (calibr*[Title/Abstract] OR quantif*[Title/ Abstract] OR normali*[Title/Abstract] OR standardi*[Title/Abstract]) AND (brain*[Title/Abstract] OR head*[Title/Abstract] OR cerebr* [Title/Abstract]) |
| Web of Science | (TS = (((magnetic W/3 resonan*) OR t1 OR t1w OR mr OR mri) AND imag* AND intensit* AND (calibr* OR quantif* OR normali* OR standardi*) AND (brain* OR head* OR cerebr*))) OR (AB = (((magnetic W/3 resonan*) OR t1 OR t1w OR mr OR mri) AND imag* AND intensit* AND (calibr* OR quantif* OR normali* OR standardi*) AND (brain* OR head* OR cerebr*))) OR (AK = (((magnetic W/3 resonan*) OR t1 OR t1w OR mr OR mri) AND imag* AND intensit* AND (calibr* OR quantif* OR normali* OR standardi*) AND (brain* OR head* OR cerebr*))) OR (KP = (((magnetic W/3 resonan*) OR t1 OR t1w OR mr OR mri) AND imag* AND intensit* AND (calibr* OR quantif* OR normali* OR standardi*) AND (brain* OR head* OR cerebr*))) |

Table 1 provides a detailed overview of the search settings applied in each database during the literature search. Scopus: TITLE-ABS-KEY indicates that any of the words mentioned thereafter should be included in the title, abstract, or keywords; W/3 indicates that the two words before and after should be located within a 3-word span; OR and AND indicate logical operators; * indicates that the word can have further characters in that specific place. PubMed: every word is followed by [Title/Abstract], which indicates that the word has to appear in the title or the abstract; OR and AND and * have the same role as in Scopus; Web of Science: search settings are defined separately for the title, the abstract, the author keywords, and the keywords plus field. Abbreviations: ABS/AB: abstract; AK: author keywords; KP: keyword plus field; TS: title.

- (C) Comparator: As a comparator, we considered other scaling methods (e.g., based on reference regions in the brain), quantitative imaging, MRI measures related to tissue microstructure, or histological examinations.

- (O) Outcome: Relation to quantitative imaging and MRI measures related to tissue microstructure. Relation to histology. Analysis of magnetic resonance imaging (MRI) intensity variance if comparison to another method was available. Relation to clinical or demographic data, if comparison to another method, to quantitative imaging or MRI measures related to tissue microstructure (such as magnetization transfer ratio), or to histology was available.

Hence, studies were included if they met the following inclusion criteria: human study; conventional T1w or T2w brain MRI; intensity scaling avoiding cerebral reference regions; comparison of scaled intensities with another method, quantitative imaging or MRI measures related to tissue microstructure, or histology; description of scaling method; accessible scientific paper; extractable results. Studies were screened for inclusion by TW and MM, and disagreements were solved through discussion. Risk of Bias was assessed with a customized tool based on the tools provided by Cochrane, QUADAS, and NIH. Further details on the search strategy, selection process, risk of bias assessment, and results reporting methods are presented in S1 File.

# 3 Results

## 3.1 Study selection and flow diagram

A detailed PRISMA flow diagram of the selection process is depicted in Fig 1. In total, 3825 studies were included in the title and abstract screening, and 24 finally met the inclusion

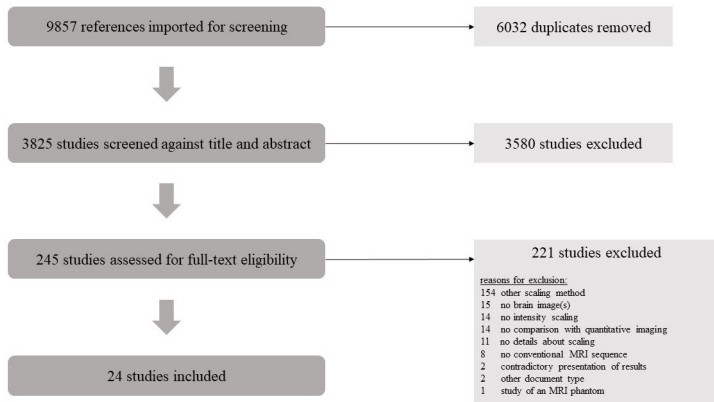

**Fig 1. PRISMA flow chart.** The flow chart shows each step of the literature screening and the number of studies excluded in each step. After screening, 24 studies remained and were included in this systematic review.

criteria. Details on exclusions are given in S1 File. The results of Risk of Bias assessment are presented in Fig 2.

## 3.2 Study characteristics

The characteristics of the 24 included studies are summarized in Table 2. Most studies focused on the scaling of T1w images (including T1w/T2w-ratios). The studies on T1w/T2w-ratios calculated the ratio either without or with scaling of the T1w or T2w images beforehand. We will refer to these approaches as either "unscaled" or "scaled" T1w/T2w-ratio. We refer to "unscaled" ratios whenever image intensities have not been scaled. However, images used to calculate the ratio may have undergone bias field correction as in most studies using T1w/T2w-ratios (Table 3 in S1 File). Only one study investigated the scaling of T2w images exclusively [17] and two studies the scaling of independent T2w and T1w images [18, 19]. Given the low number of studies and the heterogenous methodology, the results on scaling T2w images are included in the tables but will not be considered further. The included studies largely focused on MR images of healthy people but also included images of people with MS (also post-mortem), people with monophasic demyelinating disorders, people with Alzheimer's disease, people with lower-grade glioma, people with Glioblastoma, and children born preterm. The scaling methods applied in the included studies also differed. In total, seven different scaling methods were applied. Most studies implemented the method proposed by Ganzetti et al. [14], which uses the median intensity values of extracerebral reference regions (eyeball and temporal muscle) to scale T1w and T2w image intensities before calculating the T1w/T2w-ratio:

$$sI(Ganzetti) = \frac{X_R - Y_R}{X_S - Y_S} * I + \frac{X_S Y_R - X_R Y_S}{X_S - Y_S}$$

with X: median (temporal muscle), Y: median (eyeball), R: template image, S: subject image. Some studies also calculated T1w/T2w-ratio, but they did not scale T1w and T2w images beforehand. The studies by Brown et al. [20] and Gilmore et al. [18] scaled intensities by dividing the conventional MR-image intensities by the median intensity of orbital fat and by the mean intensity of subcutaneous fat, respectively. Soun et al. [21] followed the method of

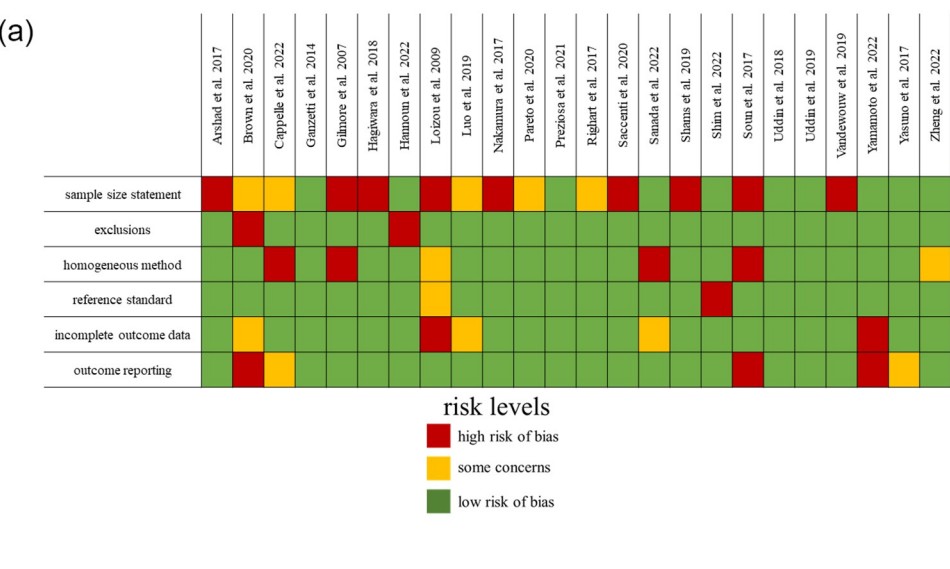

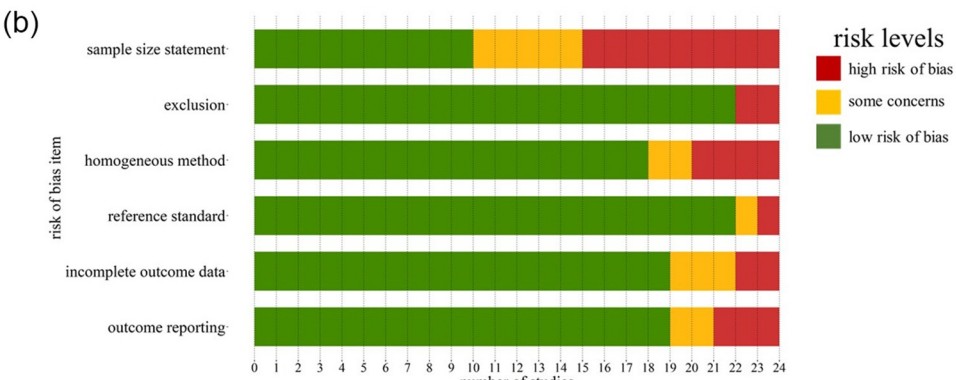

**Fig 2. Risk of Bias assessment results.** The Risk of Bias assessment results are presented in detail in (A), and the number of studies per rating category for each risk of bias item is depicted in (B). In both figures, green fields indicate a low risk of bias, yellow fields indicate some concerns, and red fields indicate a high risk of bias.

Ganzetti et al. [14] but they used the masseter muscle instead of the temporal muscle. The study by Loizou et al. [17] used the matlab function 'gscale' and histogram normalization.

## 3.3 Results of individual studies and syntheses

**3.3.1 Variance reduction.** Four studies investigated variance reduction; an overview of the results is provided in Table 3. In the study by Brown et al. [20], inter-scan variance of T1w images was reduced through scaling with orbital fat but less so when compared to scaling according to normal-appearing white matter (NAWM) through the "White-Stripe" method [7]. The two studies using scaled T1w/T2w-ratios also showed a considerable reduction of inter-scan variance as statistically significant differences between different datasets vanished when scaled T1w/T2w-ratios were used instead of unscaled T1w/T2w-ratios [14]; further, coefficients of variance were reduced for T1w/T2w-ratios and even more so for T1w/FLAIR-ratios by three different scaling methods based on histogram calibration using extracerebral tissue [22].

**Table 2. Study characteristics.**

| Study | Data Source | #Subjects | Age | Health Status / Disorder | Imaging Modalities | Reference Region(s) | Scaling Method |
|---|---|---|---|---|---|---|---|
| Arshad et al. 2017 [23] | In-house | 20 | Mean = 45.9 years | HC | T1w, T2w, MWF, geomT2IEW | eyeballs, temporal muscle | sI(Ganzetti) |
| Brown et al. 2020 [20] | In-house | 179 | 5-20 years | HC, MS, monophasic demyelinating disorders | T1w, MTR | orbital fat | sI(Brown) |
| Cappelle et al. 2022 [22] | In-house | 207 | Mean = 40.65 years | MS | T1w, T2w, FLAIR, MTR | eyeballs, temporal muscle | sI(Ganzetti) (T1w/T2w and T1w/FLAIR) |
| Ganzetti et al. 2014 [14] | (a) (b) | 63 | Mean = 31.97 years | HC | T1w, T2w, FLAIR, MTR, FA | eyeballs, temporal muscle | sI(Ganzetti) |
| Gilmore et al. 2007 [18] | In-house | 47 | Mean = 39.7 weeks gestational age | HC | T1w, T2w, DWI (FA, MD) | subcutaneous fat | sI(Gilmore) |
| Hagiwara et al. 2018 [24] | In-house | 20 | Mean = 55.3 years | HC | SyMRI Data (T1w, T2w, MVF, MTsat) | eyeballs, temporal muscle | sI(Ganzetti) |
| Hannoun et al. 2022 [25] | In-house | 52 | Mean = 36.1 years | HC, MS | T1w, T2w, DWI | eyeballs, temporal muscle | sI(Ganzetti) |
| Loizou et al. 2009 [17] | In-house | 32 | Mean = 30.75 years | HC, MS | T2w | CSF, air from sinuses | Matlab function gscale, Histogram Normalization |
| Luo et al. 2019 [26] | (c) | 252 | Mean = 73.76 years | HC, preclinical AD, prodromal AD, AD dementia | T1w, T2w, FLAIR, PET imaging | eyeballs, temporal muscle | sI(Ganzetti) |
| Nakamura et al. 2017 [27] | In-house | 6 | Mean = 62.5 years | MS (post-mortem) | T1w, T2w, MTR | | T1w/T2w-ratio |
| Pareto et al. 2020 [28] | In-house | 22 | Mean = 41.13 years | MS | T1w, T2w, MTR | | T1w/T2w-ratio |
| Preziosa et al. 2021 [45] | In-house | 25 | Mean = 67.0 years | HC, MS | T1w, T2w, FLAIR (histology) | eyeballs, temporal muscle | sI(Ganzetti) |
| Righart et al. 2017 [46] | In-house | In-vivo: 248 Post-mortem: 9 | Mean(in-vivo) = 30.8 years Mean(post-mortem) = 65.9 years | HC, MS, MS (post-mortem) | T1w, T2w, FLAIR (histology) | | T1w/T2w-ratio |
| Saccenti et al. 2020 [29] | In-house | 21 | Mean = 37.9 years | MS | SyMRI Data (T1w, T2w, MVF, MTsat) DWI | eyeballs, temporal muscle | sI(Ganzetti) |
| Sanada et al. 2022 [30] | (d) | 163 | | Lower-grade Glioma | T1w, T2w, T1-relaxometry, T2-relaxometry | eyeballs, temporal muscle | sI(Ganzetti) |
| Shams et al. 2019 [31] | In-house | 17 | Mean = 24.7 years | HC | T1w, T2w, T1-relaxometry | | T1w/T2w-ratio |
| Shim et al. 2022 [32] | In-house | 10 | Mean = 32.7 years | HC | T1w, T2*w, T1-relaxometry, T2-relaxometry | | T1w/T2*w-ratio |
| Soun et al. 2017 [21] | In-house | 10 | Mean = 38.7 weeks gestational age | HC | T1w, T2w, FLAIR, DWI | eyeballs, masseter muscle | sI(Soun) |
| Uddin et al. 2018 [19] | In-house | 10 | Mean = 57.1 years | MS | T1w, T2w, GRASE (MWF) | eyeballs, temporal muscle | sI(Ganzetti) |
| Uddin et al. 2019 [33] | In-house | 31 | Mean = 29.6 years | HC | T1w, GRASE (T2w, MWF), DWI | eyeballs, temporal muscle | sI(Ganzetti) |
| Vandewouw et al. 2019 [34] | In-house | 56 | Mean = 4.3 years | 4-year-old children (born very preterm, born full term) | T1w, T2w, MTR | | T1w/T2w-ratio |
| Yamamoto et al. 2022 [35] | In-house | 34 | Mean = 63.5 years | Glioblastoma | T1w, T2w, T1-relaxometry, T2-relaxometry, [11C]Met-PET | eyeballs, temporal muscle | sI(Ganzetti) |
| Yasuno et al. 2017 [36] | In-house | 38 | Mean = 70.4 years | HC | T1w, T2w, PET imaging | eyeballs, temporal muscle | sI(Ganzetti) |

*(Continued)*

**Table 2.** (Continued)

| Study | Data Source | #Subjects | Age | Health Status / Disorder | Imaging Modalities | Reference Region(s) | Scaling Method |
|---|---|---|---|---|---|---|---|
| Zheng et al. 2022 [37] | In-house | 9 | Mean = 64.7 years | MS (post-mortem) | T1w, T2w, MTR (histology) | | T1w/T2w-ratio |

Abbreviations: AD: Alzheimer's disease; DWI: diffusion weighted imaging; FA: fractional anisotropy; FLAIR: fluid-attenuated inversion recovery; geomT2IEW: geometric-mean of the intra-/extracellular water T2; GRASE: gradient and spin echo; HC: healthy controls; MD: mean diffusivity; Met: methionine; MS: multiple sclerosis; MTR: magnetization transfer ratio; $MT_{sat}$: magnetization transfer saturation; MWF: myelin water fraction; MVF: myelin volume fraction; PET: positron emission tomography.

(a) IXI database of the Imperial College London (http://biomedic.doc.ic.ac.uk/brain-development/index.php?n=Main.Dataset)

(b) KIRBY21 database of the Kirby Research Center for Functional Brain Imaging in Baltimore (http://mri.kennedykrieger.org/databases.html)

(c) Alzheimer's disease Neuroimaging Initiative database (adni.loni.usc.edu)

(d) In-house & Cancer Imaging Archive/Cancer Genome Atlas low-grade glioma collection dataset (https://www.cancerimagingarchive.net/)

$sI(Brown) = \frac{I}{median(orbital\ fat)}$ with I: unscaled intensity, sI: scaled intensity

$sI(Ganzetti) = \frac{X_R - Y_R}{X_S - Y_S} * I + \frac{X_S Y_R - X_R Y_S}{X_S - Y_S}$ with X: median(temporal muscle), Y: median(eyeball), R: template, S: subject

$sI(Gilmore) = \frac{I}{mean(subcutaneous\ fat)}$ with I: unscaled intensity, sI: scaled intensity

$\mathbf{sI(Soun)} = \frac{X_R - Y_R}{X_S - Y_S} * I + \frac{X_S Y_R - X_R Y_S}{X_S - Y_S}$ with X: mean(masseter muscle), Y: mean(eyeball), R: template, S: subject

**3.3.2 Relation to quantitative imaging and MRI measures related to tissue microstructure.** Twenty studies investigated relationships between scaled images and quantitative imaging [14, 18–37]. In total, scaled images were compared to 16 quantitative imaging methods (Fig 3). Detailed results summaries are shown in Table 4 (scaled T1w images, scaled T2w images, unscaled T1w/T2w-ratio) and Table 5 (scaled T1w/T2w-ratio). Although we are aware that they are not fully quantitative, we refer to the "MRI measures related to tissue microstructure", such as magnetization transfer ratio (MTR) or diffusion-weighted imaging (DWI), as "quantitative imaging" for readability.

*Scaled T1w images.* The strongest correlations were found with magnetization transfer ratio (MTR), which has been found to be sensitive to myelin, tissue integrity, and axonal counts [38–41]. Brown et al. [20] reported a strong positive relation ($R^2 = 0.63$) with MTR across the brain. Statistically significant positive correlations were also observed with myelin water fraction (MWF), a histologically validated myelin marker [40, 42–44], in subcortical grey matter (GM) regions of interest (ROIs) ($R^2 = 0.15$, $p<0.05$) but not in white matter (WM) ROIs [19]. The correlations with DWI metrics, known to be sensitive to microstructural changes, were statistically insignificant, except for the correlation between mean diffusivity (MD) and scaled T1w image intensities in the 'cortical splenium' ($R^2 = 0.16$, $p<0.01$) [18].

*Unscaled T1w/T2w-ratio.* In three studies, unscaled T1w/T2w-ratios were compared to MTR values, and they all found statistically significant positive correlations. Statistically significant positive correlations of different strengths were observed in WM ($R^2 = 0.33$, $p<0.001$), subcortical GM ($R^2 = 0.77$, $p<0.001$), and cortical GM ($R^2 = 0.25$, $p<0.001$) [34], in NAWM ($R^2 = 0.63$, $p<0.01$), normal-appearing grey matter (NAGM) ($R^2 = 0.50$, $p<0.05$), MS-lesions ($R^2 = 0.70$, $p<0.01$), and cerebrospinal fluid (CSF) ($R^2 = 0.63$, $p<0.05$) [28], and in cortical regions ($R^2 = 0.19$, $p<0.01$) [27]. In two studies, strong correlations were reported with T1 relaxation rate (R1) in cortical regions [31] ($R^2 = 0.8$) and with relaxometry-based "q-ratio" values (q-ratio = R1/T2*) in WM ($R^2 = 0.48$, $p<0.001$) as well as in GM ($R^2 = 0.82$, $p<0.001$) [32].

**Table 3. Variance reduction results.**

| Reference Region | Images | Comparative Method | Variance Assessment | ROI | before Scaling | after Scaling | after Comparative Method | Removed Variance | Scaling equal to or better than Comparative Method |
|---|---|---|---|---|---|---|---|---|---|
| | | | | | **Brown et al. 2020** | | | | |
| orbital fat | scaled T1w | White-Stripe | visual histogram inspection | Brain | large variability | low variability | low variability | ✔ | ✔ |
| | | | | | 2 peaks for WM | 1 peak for WM | 1 peak for WM | ✔ | ✔ |
| | | | | | 2 peaks for GM | 1 peak for GM | 1 peak for GM | ✔ | ✔ |
| | | | intensity range ±estimation error[a] | WM | 4.5 ±0.3 | 2 ±0.3 | 1 ±0.3 | ✔ | ✕ |
| | | | | GM | 3.5 ±0.3 | 2 ±0.3 | 0.5 ±0.3 | ✔ | ✕ |
| | | | | | **Cappelle et al. 2022** | | | | |
| eyeballs, temporal muscle | scaled T1w/T2w-ratio | nonlinear histogram calibration with generic template | Coefficient of Variance (%) | NAWM | 12.0 | 13.2 | 12.1 | ✕ | ✕ |
| | | | | NAGM | 15.1 | 13.0 | 11.7 | ✔ | ✕ |
| | | | | MS Lesions | 23.7 | 20.0 | 13.5 | ✔ | ✕ |
| | | | | Corpus Callosum | 18.5 | 18.2[#] | 13.0[*+] | ✔ | ✕ |
| | | | | Thalamus | 13.7 | 13.6 | 11.0 | ✔ | ✕ |
| | | nonlinear histogram calibration with subject template | Coefficient of Variance (%) | NAWM | 12.0 | 13.2 | 9.7 | ✕ | ✕ |
| | | | | NAGM | 15.1 | 13.0 | 13.0 | ✔ | ✔ |
| | | | | MS Lesions | 23.7 | 20.0 | 17.3 | ✔ | ✕ |
| | | | | Corpus Callosum | 18.5 | 18.2 | 13.2 | ✔ | ✕ |
| | | | | Thalamus | 13.7 | 13.6 | 10.6 | ✔ | ✕ |
| | scaled T1w/ FLAIR-ratio | nonlinear histogram calibration with generic template | Coefficient of Variance (%) | NAWM | 24.0 | 19.1[#] | 9.5[*+] | ✔ | ✕ |
| | | | | NAGM | 26.1 | 17.0[*] | 12.0[*] | ✔ | ✕ |
| | | | | MS Lesions | 27.5 | 23.0[#] | 9.7[*+§] | ✔ | ✕ |
| | | | | Corpus Callosum | 28.6 | 24.8[#] | 12.4[*+§] | ✔ | ✕ |
| | | | | Thalamus | 26.5 | 23.4[#] | 11.7[*+] | ✔ | ✕ |
| | | nonlinear histogram calibration with subject template | Coefficient of Variance (%) | NAWM | 24.0 | 19.1 | 13.8[*] | ✔ | ✕ |
| | | | | NAGM | 26.1 | 17.0[*] | 15.1[*] | ✔ | ✕ |
| | | | | MS Lesions | 27.5 | 23.0 | 22.0[#] | ✔ | ✕ |
| | | | | Corpus Callosum | 28.6 | 24.8 | 18.6[*#] | ✔ | ✕ |
| | | | | Thalamus | 26.5 | 23.4[§] | 16.5[*+] | ✔ | ✕ |
| | | | | | **Ganzetti et al. 2014** | | | | |

(*Continued*)

**Table 3.** (Continued)

| Reference Region | Images | Comparative Method | Variance Assessment | ROI | before Scaling | after Scaling | after Comparative Method | Removed Variance | Scaling equal to or better than Comparative Method |
|---|---|---|---|---|---|---|---|---|---|
| eyeballs, temporal muscle | scaled T1w/T2w-ratio | unscaled T1w/T2w-ratio | visual histogram inspection | Brain | different intensity scales | similar intensity scales | (see before scaling) | ✓ | ✓ |
| | | | | | large between-dataset and inter-subject variability | reduced between-dataset and inter-subject variability | (see before scaling) | ✓ | ✓ |
| | | | mean ±sd | WM | IXI 1.5T: 3.44 ±0.31 | IXI 1.5T: 2.11 ±0.15 | (see before scaling) | ✓ | ✓ |
| | | | | | IXI 3T: 4.52 ±0.82 | IXI 3T: 2.04 ±0.10 | (see before scaling) | ✓ | ✓ |
| | | | | | KIRBY21 3T: 28.49 ±4.63 | KIRBY21 3T: 2.07 ±0.14 | (see before scaling) | ✓ | ✓ |
| | | | ANOVA (difference between datasets) | WM | $p < 0.001$ | $p = 0.2236$ | (see before scaling) | ✓ | ✓ |

**Loizou et al. 2009**

(*Continued*)

**Table 3.** (Continued)

| Reference Region | Images | Comparative Method | Variance Assessment | ROI | before Scaling | after Scaling | after Comparative Method | Removed Variance | Scaling equal to or better than Comparative Method |
|---|---|---|---|---|---|---|---|---|---|
| sinus air, CSF | scaled T2w (intensity scaling) | contrast stretch and normalization | Kullback Leibler Divergence Distance[(b)] (mean ±sd) | Brain | intra-scan: 0.336 ±0.130 | intra-scan: 0.112 ±0.00 | intra-scan: 0.156 ±0.01 | ✓ | ✓ |
| | | | | | inter-scan: 0.406 ±0.135 | inter-scan: 0.250 ±0.017 | inter-scan: 0.247 ±0.015 | ✓ | ✓ |
| | | histogram stretching | | | intra-scan: 0.336 ±0.130 | intra-scan: 0.112 ±0.00 | intra-scan: 0.178 ±0.020 | ✓ | ✓ |
| | | | | | inter-scan: 0.406 ±0.135 | inter-scan: 0.250 ±0.017 | inter-scan: 0.298 ±0.023 | ✓ | ✓ |
| | | gaussian kernel normalization | | | intra-scan: 0.336 ±0.130 | intra-scan: 0.112 ±0.00 | intra-scan: 0.165 ±0.00 | ✓ | ✓ |
| | | | | | inter-scan: 0.406 ±0.135 | inter-scan: 0.250 ±0.017 | inter-scan: 0.290 ±0.010 | ✓ | ✓ |
| | | histogram equalization | | | intra-scan: 0.336 ±0.130 | intra-scan: 0.112 ±0.00 | intra-scan: 0.185 ±0.010 | ✓ | ✓ |
| | | | | | inter-scan: 0.406 ±0.135 | inter-scan: 0.250 ±0.017 | inter-scan: 0.287 ±0.006 | ✓ | ✓ |
| | scaled T2w (histogram normalization) | contrast stretch and normalization | Kullback Leibler Divergence Distance[(b)] (mean ±sd) | Brain | intra-scan: 0.336 ±0.130 | intra-scan: 0.099 ±0.00 | intra-scan: 0.156 ±0.01 | ✓ | ✓ |
| | | | | | inter-scan: 0.406 ±0.135 | inter-scan: 0.142 ±0.010 | inter-scan: 0.247 ±0.015 | ✓ | ✓ |
| | | histogram stretching | | | intra-scan: 0.336 ±0.130 | intra-scan: 0.099 ±0.00 | intra-scan: 0.178 ±0.020 | ✓ | ✓ |
| | | | | | inter-scan: 0.406 ±0.135 | inter-scan: 0.142 ±0.010 | inter-scan: 0.298 ±0.023 | ✓ | ✓ |
| | | gaussian kernel normalization | | | intra-scan: 0.336 ±0.130 | intra-scan: 0.099 ±0.00 | intra-scan: 0.165 ±0.00 | ✓ | ✓ |
| | | | | | inter-scan: 0.406 ±0.135 | inter-scan: 0.142 ±0.010 | inter-scan: 0.290 ±0.010 | ✓ | ✓ |
| | | histogram equalization | | | intra-scan: 0.336 ±0.130 | intra-scan: 0.099 ±0.00 | intra-scan: 0.185 ±0.010 | ✓ | ✓ |
| | | | | | inter-scan: 0.406 ±0.135 | inter-scan: 0.142 ±0.010 | inter-scan: 0.287 ±0.006 | ✓ | ✓ |

Table 3 summarizes data on variance reduction for eligible studies. For each result, we indicated if the scaling method in question removed variance and whether it removed more or less variance than the comparative method. Abbreviations: ANOVA: analysis of variance; CSF: cerebrospinal fluid; FLAIR: fluid attenuated inversion recovery; GM: gray matter; MS: multiple sclerosis; NAWM/NAGM: normal-appearing white/gray matter; NLG: nonlinear histogram calibration with generic template; NLS: nonlinear histogram calibration with subject template; Sd: standard deviation; ROI: region(s) of interest; WM: white matter.

Statistically significant difference with unscaled ratio: *, statistically significant difference with scaled ratio (Ganzetti): +, statistically significant difference with NLG-scaled: #; statistically significant difference with NLS-scaled: §.

[(a)]: We visually estimated the intensity ranges (given in standardized intensity units) from the intensity histograms because no value was provided by the authors. The values should, therefore, be considered with an error of +/-0.3 due to possible inaccuracies in the extraction process.

[(b)]: intra-scan Kullback Leibler Divergence (KLD) Distance was calculated between different slices from the same scan, and inter-scan KLD Distance was calculated between corresponding slices from different scans (i.e., corresponding slices of original images and scaled images)

*Scaled T1w/T2w-ratio*. Two studies assessed the relation to MTR values only qualitatively. A positive relation of MTR values with scaled T1w/T2w-ratios [14, 22] and scaled T1w/FLAIR-ratios [22] was described.

Correlations with DWI were weak and inconsistent. No congruent results were observed with radial diffusivity (RD) as correlations were reported to be statistically significant [25] or

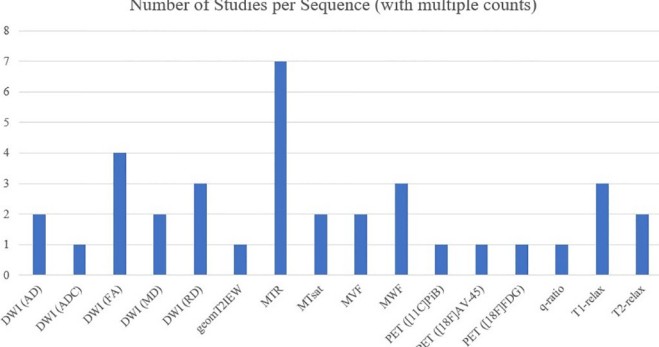

**Fig 3. Availability of quantitative imaging and MRI measures related to tissue microstructure.** This histogram shows the number of studies having investigated each quantitative imaging technique (with multiple counts regarding studies because they might have compared multiple quantitative sequences). Abbreviations: AD: axial diffusivity; ADC: apparent diffusion coefficient; AV: amyvid; DWI: diffusion weighted imaging; FA: fractional anisotropy; FDG: fluorodeoxyglucose; MD: mean diffusivity; MTR: magnetization transfer ratio; $MT_{sat}$: magnetization transfer saturation; MVF: myelin volume fraction; MWF: myelin water fraction; PET: positron emission tomography; PiB: Pittsburgh compound B; RD: radial diffusivity.

insignificant [29], [33] in WM regions; in MS WM lesions correlations were statistically significant (Spearman rho = -0.58, p<0.001) [29]; in subcortical GM, no statistically significant correlations were observed [33]. Correlations with DWI fractional anisotropy (FA) were also inconsistent since they were statistically significant [25] or insignificant [33] in WM, and statistically significant correlations were found in GM [14, 33]. Likewise, correlations with axial diffusivity (AD) were statistically significant in subcortical GM ($R^2$ = 0.12, p<0.000001 Bonferroni corrected) [33] but, in WM, statistically insignificant, negative, or positive [25, 33]. No statistically significant correlation was observed with MD [33], and a statistically significant positive correlation with the apparent diffusion coefficient (ADC) was found in the posterior limb of the internal capsule and in the optic radiation ($R^2$ = 0.96, p<0.001) [21].

Surprisingly, three studies found differing results on correlations with MWF and reported statistically significant negative ($R^2$ = 0.07, p<0.05) [23] or positive ($R^2$ = 0.06–0.11) [33] correlations in healthy people as well as statistically significant positive and insignificant correlations [19] in people with MS.

Correlations with multi-echo T2-based geometric-mean of the intra-/extracellular water T2 (geomT2IEW) values, which are supposed to be sensitive to the diameter of axons, were statistically significant (negative) in WM of healthy people ($R^2$ = 0.29, p<0.05) [23].

Negative correlations of scaled T1w/T2w-ratios with T1- and T2-relaxation times were found within lower-grade glioma (T1-ralaxation: $R^2$ = 0.64; T2-relaxation: $R^2$ = 0.59) [30] and within non-contrast-enhancing glioblastoma lesions (T1-relaxation: $R^2$ = 0.002; T2-relaxation: $R^2$ = 0.07) [35].

Using synthetic MRI data, two studies found statistically significant positive correlations between scaled T1w/T2w-ratios and myelin volume fraction (MVF) (Spearman rho = 0.45–0.89), assumed to measure the amount of myelin per voxel, and with magnetization transfer saturation (MTsat) (Spearman rho = 0.28–0.80), assumed to be sensitive to microstructural integrity, across various brain regions [24, 29]. Notice that in these two studies, the conventional T1w and the T2w images were estimated from SyMRI multiparameter mapping, and resulting intensities were subsequently scaled and used for T1w/T2w-ratio calculation.

**Table 4. Relation between scaled T1w, scaled T2w, and unscaled T1w/T2w-ratio images and quantitative imaging or MRI measures related to tissue microstructure.**

| Reference Region(s) | Quantitative Sequence | ROI | #Scans | Results | Significance |
|---|---|---|---|---|---|
| | | **Scaled T1w** | | | |
| | | **Brown et al. 2020** | | | |
| orbital fat | MTR | brain | 650 | 0.63 (positive) | NA |
| | | **Gilmore et al. 2007** | | | |
| subcutaneous fat | FA | genu (central) | 47 | no correlation | |
| | | genu (cortical) | 47 | no correlation | |
| | | splenium (central) | 47 | no correlation | |
| | | splenium (cortical) | 47 | no correlation | |
| | | corticospinal tract (central) | 47 | no correlation | |
| | | corticospinal tract (cortical) | 47 | no correlation | |
| | MD | genu (central) | 47 | no correlation | |
| | | genu (cortical) | 47 | no correlation | |
| | | splenium (central) | 47 | no correlation | |
| | | splenium (cortical) | 47 | 0.1602 (negative) | ** |
| | | corticospinal tract (central) | 47 | no correlation | |
| | | corticospinal tract (cortical) | 47 | no correlation | |
| | | **Uddin et al. 2018** | | | |
| eyeballs, temporal muscle | MWF | WM | 10 | 0.006 (positive) | |
| | | subcortical GM | 10 | 0.15 (positive) | * |
| | | brain | 10 | 0.056 (positive) | ** |
| | | **Scaled T2w** | | | |
| | | **Gilmore et al. 2007** | | | |
| CSF | FA | genu (central) | 47 | negative | * |
| | | genu (cortical) | 47 | negative | * |
| | | splenium (central) | 47 | negative | * |
| | | splenium (cortical) | 47 | negative | * |
| | | corticospinal tract (central) | 47 | negative | * |
| | | corticospinal tract (cortical) | 47 | negative | * |
| | MD | genu (central) | 47 | positive | * |
| | | genu (cortical) | 47 | positive | * |
| | | splenium (central) | 47 | positive | * |
| | | splenium (cortical) | 47 | positive | * |
| | | corticospinal tract (central) | 47 | positive | * |
| | | corticospinal tract (cortical) | 47 | positive | * |
| | | **Uddin et al. 2018** | | | |
| eyeballs, temporal muscle | MWF | WM | 10 | 0.048 (positive) | * |
| | | subcortical GM | 10 | 0.23 (positive) | ** |
| | | brain | 10 | 0.12 (positive) | *** |
| | | **Unscaled T1w/T2w** | | | |
| | | **Nakamura et al. 2017** | | | |
| | MTR | cortical regions | 6 | 0.194 (positive) | ** |
| | | **Shams et al. 2019** | | | |
| | T1 relaxometry (R1) | cortical regions | 17 | 0.74 (positive) | NA |
| | | **Shim et al. 2022** | | | |
| | q-ratio ($R1/T2^*$) | WM | 10 | 0.4833 (positive) | *** |
| | | GM | 10 | 0.819 (positive) | *** |

*(Continued)*

**Table 4.** (Continued)

| Reference Region(s) | Quantitative Sequence | ROI | #Scans | Results | Significance |
|---|---|---|---|---|---|
| | | **Vandewouw et al. 2019** | | | |
| | MTR | WM | 56 | 0.33 (positive) | *** |
| | | subcortical GM | 56 | 0.77 (positive) | *** |
| | | cortical GM | 56 | 0.25 (positive) | *** |
| | | **Pareto et al. 2020** | | | |
| | MTR | NAWM | 22 | 0.631 (positive) | ** |
| | | NAGM | 22 | 0.502 (positive) | * |
| | | MS-lesions | 22 | 0.699 (positive) | ** |
| | | CSF | 22 | 0.631 (positive) | * |

Table 4 summarizes relations between scaled T1w, scaled T2w, and unscaled T1w/T2w-ratio images and quantitative imaging. The "Results" column shows the quantified relation in terms of $R^2$ values derived from linear regression, with an indication of negative or positive correlation. All correlations were calculated across subjects using mean intensities from indicated regions of interest. Abbreviations: CSF: cerebrospinal fluid; FA: fractional anisotropy; GM: gray matter; MD: mean diffusivity; MS: multiple sclerosis; MTR: magnetization transfer ratio; MWF: myelin water fraction; NA: not available/applicable; NAWM/NAGM: normal-appearing; ROI: region of interest; WM: white matter.

*: $p < 0.05$,

**: $p < 0.01$,

***: $p < 0.001$

**3.3.3 Relation to clinical or demographical data.** Eight studies used scaled images to investigate group differences (e.g., healthy controls vs. people with MS) or the relation of scaled image intensities with clinical or demographical data [18, 20, 25, 26, 28, 29, 34, 45].

Scaled T1w image intensities correlated with age at onset and disease duration for monophasic acquired demyelinating syndrome and MS, similar to MTR values [20], but no coherent trend was observed for scaled T1w image intensities and DWI-metrics (FA and MD) in their correlation with gestational age [18].

Unscaled T1w/T2w-ratios and MTR values did not correlate with gestational age but with development assessment in WM, whereas, in this regard, only T1w/T2w-ratio correlated in cortical GM and MTR in subcortical GM [34]. Correlations with Expanded Disability Status Scale (EDSS) were similar for unscaled T1w/T2w-ratios and MTR values, but only unscaled T1w/T2w-ratios showed statistically significant negative correlations with disease duration within MS WM lesions [28].

Scaled T1w/T2w-ratios in WM were statistically significantly decreased in people with MS (compared to healthy people), which went along with a statistically significant decrease in FA, an increase in AD, and an increase in RD [25]. In contrast to the results on unscaled T1w/T2w-ratios [28], scaled T1w/T2w-ratios based on synthetic MRI did not statistically significantly correlate with EDSS and with disease duration [29].

**3.3.4 Relation to histology.** Four studies related scaled or unscaled T1w/T2w-ratios to histology-based measures [27, 37, 45, 46], these results are summarized in Table 5 in S1 File.

In one study, scaled T1w/T2w-ratios statistically significantly correlated with neurite density, whereas no correlation with myelin density was observed in the cortex [45]. When investigating group differences between healthy people (n = 10) and people with MS (n = 15) in the normal-appearing cortex, statistically significant differences for scaled T1w/T2w-ratios (p = 0.045) and neurite-density (p = 0.041), but not for myelin density, were found.

Three studies compared unscaled T1w/T2w-ratios to histology, all within the cortex [27, 37, 46]. Unscaled T1w/T2w-ratios in histologically confirmed demyelinated areas were statistically

**Table 5. Relation between scaled T1w/T2w-ratios and quantitative imaging or MRI measures related to tissue microstructure.**

| Reference Region (s) | Quantitative Sequence | ROI | #Scans | Results | p-Value | Comparison Metric | Voxel-level (VL) / Region-level (RL) Correlation |
|---|---|---|---|---|---|---|---|
| | | | | **Arshad et al. 2017** | | | |
| eyeballs, temporal muscle | MWF | WM | 20 | 0.068 (negative) | * | $R^2$, linear regression | RL |
| | geomT2IEW | WM | 20 | 0.292 (negative) | * | $R^2$, linear regression | RL |
| | | | | **Soun et al. 2017** | | | |
| eyeballs, masseter muscle | ADC | PLIC & optic radiation | 10 | 0.96 (positive) | *** | $R^2$, linear regression | VL |
| | | | | **Uddin et al. 2018** | | | |
| eyeballs, temporal muscle | MWF | WM | 10 | 0.00002 (positive) | | $R^2$, linear regression | RL |
| | | Subcortical GM | 10 | 0.2 (positive) | ** | $R^2$, linear regression | RL |
| | | Brain | 10 | 0.053 (positive) | ** | $R^2$, linear regression | RL |
| | | | | **Uddin et al. 2019** | | | |
| eyeballs, temporal muscle | MWF | WM | 31 | 0.061 (positive) | + | $R^2$, linear regression | RL |
| | | Subcortical GM | 31 | 0.060 (positive) | + | $R^2$, linear regression | RL |
| | | Brain | 31 | 0.106 (positive) | + | $R^2$, linear regression | RL |
| | FA | WM | 31 | 0.002 (positive) | | $R^2$, linear regression | RL |
| | | Subcortical GM | 31 | 0.082 (positive) | + | $R^2$, linear regression | RL |
| | | Brain | 31 | 0.060 (positive) | + | $R^2$, linear regression | RL |
| | AD | WM | 31 | 0.003 (negative) | | $R^2$, linear regression | RL |
| | | Subcortical GM | 31 | 0.116 (positive) | + | $R^2$, linear regression | RL |
| | | Brain | 31 | 0.021 (positive) | + | $R^2$, linear regression | RL |
| | RD | WM | 31 | 0.005 (positive) | | $R^2$, linear regression | RL |
| | | Subcortical GM | 31 | 0.012 (positive) | | $R^2$, linear regression | RL |
| | | Brain | 31 | 0.005 (positive) | | $R^2$, linear regression | RL |
| | MD | WM | 31 | 0.002 (negative) | | $R^2$, linear regression | RL |
| | | Subcortical GM | 31 | 0.023 (positive) | | $R^2$, linear regression | RL |
| | | Brain | 31 | 0.001 (negative) | | $R^2$, linear regression | RL |
| | | | | **Yamamoto et al. 2022** | | | |
| eyeballs, temporal muscle | T1 relaxometry | non-enhancing tumor | 2 | 0.002 (negative) | * | $R^2$, linear regression | VL |
| | T2 relaxometry | non-enhancing tumor | 2 | 0.067 (negative) | * | $R^2$, linear regression | VL |
| | | | | **Sanada et al. 2022** | | | |

*(Continued)*

**Table 5.** (Continued)

| Reference Region (s) | Quantitative Sequence | ROI | #Scans | Results | p-Value | Comparison Metric | Voxel-level (VL) / Region-level (RL) Correlation |
|---|---|---|---|---|---|---|---|
| eyeballs, temporal muscle | T1 relaxometry | lower-grade glioma | 8 | 0.64 (negative) | NA | $R^2$, exponential regression | VL |
| | T2 relaxometry | lower-grade glioma | 8 | 0.59 (negative) | NA | $R^2$, exponential regression | VL |
| **Hagiwara et al. 2018** | | | | | | | |
| eyeballs, temporal muscle | MVF | WM | 20 | 0.450 | *** | Spearman rho | VL |
| | | Subcortical GM | 20 | 0.690 | *** | Spearman rho | VL |
| | | Cortical GM | 20 | 0.750 | *** | Spearman rho | VL |
| | | Brain | 20 | 0.870 | *** | Spearman rho | VL |
| | MTsat | WM | 20 | 0.380 | *** | Spearman rho | VL |
| | | Subcortical GM | 20 | 0.680 | *** | Spearman rho | VL |
| | | Cortical GM | 20 | 0.540 | *** | Spearman rho | VL |
| | | Brain | 20 | 0.800 | *** | Spearman rho | VL |
| **Saccenti et al. 2020** | | | | | | | |
| eyeballs, temporal muscle | MVF | NAWM | 21 | 0.500 | *** | Spearman rho | RL |
| | | Plaque (MS) | 21 | 0.780 | *** | Spearman rho | RL |
| | | Periplaque (MS) | 21 | 0.620 | *** | Spearman rho | RL |
| | | NAWM+plaque +periplaque | 21 | 0.890 | *** | Spearman rho | RL |
| | MTsat | NAWM | 21 | 0.280 | * | Spearman rho | RL |
| | | Plaque (MS) | 21 | 0.640 | *** | Spearman rho | RL |
| | | Periplaque (MS) | 21 | 0.410 | *** | Spearman rho | RL |
| | | NAWM+plaque +periplaque | 21 | 0.800 | *** | Spearman rho | RL |
| | RD | NAWM | 21 | 0.110 | | Spearman rho | RL |
| | | Plaque (MS) | 21 | -0.580 | *** | Spearman rho | RL |
| | | Periplaque (MS) | 21 | -0.090 | | Spearman rho | RL |
| | | NAWM+plaque +periplaque | 21 | -0.660 | *** | Spearman rho | RL |
| **Yasuno et al. 2017** | | | | | | | |
| eyeballs, temporal muscle | [11C]PiB-PET | Orbital frontal cortex | 38 | 0.210 | | Spearman rho | RL |
| | | Lateral prefrontal cortex | 38 | 0.240 | | Spearman rho | RL |
| | | Medial prefrontal cortex | 38 | 0.540 | *** | Spearman rho | RL |
| | | Anterior cingulate cortex | 38 | 0.250 | | Spearman rho | RL |
| | | Medial frontal cortex | 38 | 0.490 | *** | Spearman rho | RL |
| | | Cortex | 38 | 0.450 | ** | Spearman rho | RL |
| **Hannoun et al. 2022** | | | | | | | |
| eyeballs, temporal muscle | FA | 16/24 WM ROIs | 52 | range: 0.37–0.54 | * | Spearman rho | RL |
| | AD | 8/24 WM ROIs | 52 | range: -0.51–0.50 | * | Spearman rho | RL |
| | RD | 18/24 WM ROIs | 52 | range: -0.73–(-0.38) | * | Spearman rho | RL |
| **Luo et al. 2019** | | | | | | | |

*(Continued)*

**Table 5.** (Continued)

| Reference Region (s) | Quantitative Sequence | ROI | #Scans | Results | p-Value | Comparison Metric | Voxel-level (VL) / Region-level (RL) Correlation |
|---|---|---|---|---|---|---|---|
| eyeballs, temporal muscle | [18F]FDG PET | Right inferior parietal lobule | 252 | NA | | r, partial correlation | RL |
| | | Left inferior parietal lobule | 252 | NA | | r, partial correlation | RL |
| | | Right hippocampus | 252 | 0.380 | *** | r, partial correlation | RL |
| | | Left hippocampus | 252 | 0.400 | * | r, partial correlation | RL |
| | [18F]AV-45 PET | Right inferior parietal lobule | 252 | no correlation | | r, partial correlation | RL |
| | | Left inferior parietal lobule | 252 | no correlation | | r, partial correlation | RL |
| | | Right hippocampus | 252 | 0.430 | * | r, partial correlation | RL |
| | | Left hippocampus | 252 | 0.410 | * | r, partial correlation | RL |
| **Ganzetti et al. 2014** | | | | | | | |
| eyeballs, temporal muscle | MTR | Brain | 21 | similar trend & amplitude change | NA | comparison of t-scores | RL |
| | FA | Brain | 21 | similar trend & different amplitude change | NA | comparison of t-scores | RL |

Table 5 summarizes data on relations between scaled T1w/T2w-ratio images and quantitative imaging. The "Results" column shows the quantified relation in terms of $R^2$ values derived from linear regression, $R^2$ values derived from exponential regression, Spearman rho, r from partial correlations, or comparison of t-scores, with an indication of negative or positive correlation. All correlations were calculated across subjects using mean intensities from regions of interest, indicated as region-level, or using intensities from voxels, indicated as voxel-level. Abbreviations: AD: axial diffusivity; ADC: apparent diffusion coefficient; AV: amyvid; FA: fractional anisotropy; FDG: fluorodeoxyglucose; GM: gray matter; MD: mean diffusivity; MS: multiple sclerosis; MTR: magnetization transfer ratio; MTsat: magnetization transfer saturation; MVF: myelin volume fraction; MWF: myelin water fraction; NA: not available/applicable; NAWM: normal-appearing white matter; PET: positron emission tomography; PiB: Pittsburgh compound B; PLIC: posterior limb of the internal capsule; RD: radial diffusivity; ROI: region(s) of interest; WM: white matter.

*: $p < 0.05$,

**: $p < 0.01$,

***: $p < 0.001$,

+: $p < 0.000001$ (Bonferroni corrected)

significantly lower than in myelinated regions, whereas differences in MTR values were statistically insignificant [27]. Analyzing the relations of unscaled T1w/T2w-ratios with histology within the cortex yielded statistically significant relations only between unscaled T1w/T2w-ratios and dendrite density [46].

Based on different image intensities, Zheng et al. [37] found that using unscaled T1w/T2w-ratios and MTR both yielded a high sensitivity but relatively low specificity and accuracy in detection of demyelination, whereas unscaled T2w image intensities yielded lower sensitivity but higher specificity and accuracy.

## 4 Discussion

### 4.1 Interpretation

Since conventional T1w and T2w MRI sequences are broadly available, leveraging their informative wealth beyond visual inspection and volumetry is very attractive. Applying scaling methods before analyses aims to remove the technically induced variance of intensity values while preserving the biological variance. In this systematic review, we focused on studies that avoided brain regions as reference regions because this might include a disease effect in the

scaling method (e.g., in inflammatory diseases such as MS or neurodegenerative diseases such as Alzheimer's).

**Variance reduction.** Different scaling methods have been suggested, and each of them indeed considerably reduced variance. The reduction of inter-scan variance through intensity scaling is neither surprising nor a proof of validity but a prerequisite for biologically meaningful intensity scaling. The results from Ganzetti et al. [14] showcase the potential of scaling for reducing technical variance across different datasets, and, in a later study [47], their results even indicated that biological variance is preserved. The study by Brown et al. [20] points towards better variance reduction through scaling with NAWM, however, this also comprises the possibility of removed or aggravated biological variance. Hence, variance reduction is a necessary but not a sufficient condition for evaluating intensity scaling methods. In conclusion, sparse evidence suggests that the decrease in inter-scan variance with the preservation of considerable biological variance can be achieved without specific cerebral reference regions. This variance reduction can provide many advantages in studies that need to harmonize multi-subject or multi-center images. Previous studies have investigated and highlighted the importance of intensity scaling in radiomics studies [5, 48]. Hence, radiomics studies dealing with neurodegenerative disease can aim for data harmonization without using cerebral reference regions before calculating radiomics features.

**Relation to quantitative imaging and MRI measures related to tissue microstructure.** As signal intensities of conventional T1w and T2w images are a composite measure, we did not expect to identify a single definitive (biological) driver of the signal intensity of a conventional sequence. Nonetheless, we considered it possible to observe a coherent pattern with some quantitative measure correlating more strongly to scaled conventional images than others—possibly even providing hints on the main biological drivers of signal intensities in different brain compartments. This was not the case, however. From the perspective of MRI physics, the positive correlation of T1w/T2w-ratio values with quantitative T1 or T2 parameters can be regarded as a truism. Yet, these correlations were found but not as strong and consistent as expected (range of $R^2$: 0.002–0.82). In contrast, we found robust evidence for a correlation of the T1w signal and the T1w/T2w-ratios with MTR (range of $R^2$: 0.25–0.77), which has been validated with histology and was found to correlate with myelin, tissue integrity, and axonal counts [38–41]. Surprisingly, only weak (range of $R^2$: 0.05–0.23) or no correlations were reported between scaled T1w, scaled T2w, or T1w/T2w-ratio intensity values and MWF [18, 19] even in WM, although MWF has histologically been validated as a marker of myelin [40, 42–44]. Moreover, we found no evidence for the intuitive approach to validate scaling of T1w (or T2w) image intensities through comparison with quantitative T1 (or T2) mapping. We only found studies that investigated correlations between quantitative T1 (or T2) mapping and T1w/T2w-ratios (and not with intensity-scaled T1w or T2w images).

**Relation to clinical or demographical data.** To explore the clinical correlates that could potentially be associated with intensity changes in scaled conventional T1w and T2w MR images, we summarized the data from 8 studies [18, 20, 25, 26, 28, 29, 34, 45]. Overall, statistically significant correlations with clinical data have been reported, indicating that scaled images can indeed comprise valuable information on brain pathology or development. In line with the results from the direct comparison between scaled and quantitative images, analyses with MTR and with scaled image intensities yielded similar results. However, analyses were conducted for different ROIs within the brain, which hampers the generalizability of results and identification of possible biological proxies.

**Relation to histology.** No results were available for WM, which contains considerably more axons and myelin than GM. Regarding GM, four studies [27, 37, 45, 46] provided valuable results on the biological information represented by scaled or unscaled T1w/T2w-ratios.

Although it has been assumed to be a proxy for myelin in multiple studies, the studies included in this review do not support this hypothesis within the cerebral cortex. This has also previously been commented on [49], and further factors other than myelin seem to interfere. Only one of 4 studies showed a statistically significant relation between myelin and T1w/T2w-ratios in the cortex, but only when comparing demyelinated to non-demyelinated areas [27]. T1w/T2w-ratios seem to be sensitive to various biological properties since they were statistically significantly related to neurite density [45] and to dendrite density [46]. Therefore, and also in line with the relations to quantitative imaging, we believe that the T1w/T2w-ratio requires a new interpretation, similar to the signal of a new MRI sequence. The concept of T1w/T2w-ratio representing a measure of tissue integrity is tempting, but this needs further validation and distinction. In addition, open questions remain, such as the meaning of T1w/T2w-ratios in cases where T1w and T2w image intensities behave similarly (e.g., T1w-high and T2w-high intensities in extracellular meta-hemoglobin from bleedings).

## 4.2 Limitations

Due to the very heterogeneous nature of scaling and reference methods applied in the included studies, comparing results and combining findings was only possible in descriptive ways. Heterogeneity originated from multiple aspects: 1) studies applied different methods for scaling of conventional T1w and T2w MR images, even for calculating T1w/T2w-ratios different approaches were used; 2) reference methods consisted of either many different quantitative imaging techniques, other scaling methods, or various histology analyses; 3) the ROIs that were investigated varied strongly across studies (tumor, cortex, MS-lesions, WM, subcortical GM, and others); 4) different strategies for (statistical) analyses were applied; 5) most sample sizes were small. Consequently, it was impossible to aggregate results through a meta-analysis. We tried to collect and present results as homogeneously as possible. However, some methodological differences regarding correlation analyses exist. Some studies calculated correlations with quantitative MRI or histology across subjects and regions, and some calculated the correlations across subjects in each region individually. This seemingly little difference may have influenced the results, however. First, including more regions, and, hence, more data points, may lead to increased statistical power; second, if the same subject contributes to more than one data point, the assumptions of Pearson's correlation are potentially violated.

Finally, the approach of intensity scaling with an extracerebral reference region per se is not without limitations. Even this way, interference with biological effects cannot be ruled out with certainty. For example, considerable changes in the temporal muscle have consistently been found in patients with glioma [50]. Moreover, if established, a scaling method may work reliably only with MRI data acquired at the same scanner with the same protocol.

## 4.3 Conclusion

We believe it is too early to give definite and detailed recommendations on scaling conventional T1w and T2w MR images. However, the best available evidence is on T1w/T2w ratio imaging. In this regard, we lean towards scaling images before T1w/T2w-ratio calculation in heterogeneous datasets to increase image comparability but not necessarily in homogeneous datasets (e.g., if images were acquired on the same scanner with the same protocol). Regarding scaling methods in general (and before calculating T1w/T2w-ratios), cerebral reference regions can be appropriate if not the whole brain is affected by the state under examination. If the whole brain is likely to be affected, following the suggestions by Ganzetti et al. [14] and using extracerebral reference regions presents an attractive alternative. However, we must emphasize that even extracerebral structures can be affected by the state under examination, as elaborated

for glioblastoma, multiple sclerosis, and aging in previous publications [51–53], and should therefore be selected with care. Future studies should analyze the effect of the biological factors, such as age or the disease under examination, on the reference region before applying intensity scaling. Scaling methods should be evaluated according to available evidence. We regard correlation with histology or quantitative MRI (or MRI measures related to tissue microstructure) as most desirable, followed by preservation or demonstration of known associations with clinical parameters such as group differences or correlations with clinical scores. In contrast, evidence based on mere variance reduction is precarious as it can go along with removing biologically important variance. In addition, even though Ganzetti et al. [14] reported bias correction and scaling before T1w/T2w-ratio calculation to be helpful for their datasets, this is not necessarily the case for other datasets from different centers and with other MRI scanners, sequence settings, and protocols [54].

In summary, intensity scaling of conventional T1w and T2w MR images without cerebral reference regions is feasible, leading to variance reduction while possibly preserving biologically meaningful information, with the potential to allow for signal quantification as a compound measure of tissue integrity. However, the overall level of evidence is low, with numerous open methodological questions.

## Supporting information

**S1 Checklist. PRISMA 2020 checklist.** Checklist with items for the PRISMA guideline for reporting systematic reviews.
(PDF)

**S1 File. Supplementary material.** Further information on the systematic review process and the included studies.
(DOCX)

## Author Contributions

**Conceptualization:** Tun Wiltgen, Cuici Voon, Mark Mühlau.

**Data curation:** Tun Wiltgen, Mark Mühlau.

**Funding acquisition:** Koen Van Leemput.

**Investigation:** Tun Wiltgen, Mark Mühlau.

**Methodology:** Tun Wiltgen, Cuici Voon, Mark Mühlau.

**Project administration:** Tun Wiltgen, Mark Mühlau.

**Supervision:** Koen Van Leemput, Benedikt Wiestler, Mark Mühlau.

**Writing – original draft:** Tun Wiltgen, Mark Mühlau.

**Writing – review & editing:** Tun Wiltgen, Koen Van Leemput, Benedikt Wiestler.

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
