## [Decision Letter · Decision Letter 0]

8 Jan 2024

PONE-D-23-37143Intensity scaling of conventional brain magnetic resonance images avoiding cerebral reference regions: A systematic reviewPLOS ONE

Dear Dr. Wiltgen,

Thank you for submitting your manuscript to PLOS ONE. After careful consideration, we feel that it has merit but does not fully meet PLOS ONE’s publication criteria as it currently stands. Therefore, we invite you to submit a revised version of the manuscript that addresses the points raised during the review process.  Reviewer 2 has raised significant concerns. It is recommended to incorporate additional analyses and clarifications to get the manuscript accepted.  

We look forward to receiving your revised manuscript.

Kind regards,

Md Nasir Uddin, PhD

Academic Editor

PLOS ONE

Journal Requirements:

"This study has received funding by a research grant of the National Institutes of Health (grant 1R01NS112161-01) and by a research grant of the German Research Foundation, DFG Priority Programme 2177, Radiomics: Next Generation of Biomedical Imaging (grant 428223038)"

Reviewers' comments:

Reviewer's Responses to Questions

**Comments to the Author**

1. Is the manuscript technically sound, and do the data support the conclusions?

Reviewer #1: Yes

Reviewer #2: Yes

2. Has the statistical analysis been performed appropriately and rigorously? 

Reviewer #1: Yes

Reviewer #2: Yes

3. Have the authors made all data underlying the findings in their manuscript fully available?

Reviewer #1: Yes

Reviewer #2: Yes

4. Is the manuscript presented in an intelligible fashion and written in standard English?

Reviewer #1: Yes

Reviewer #2: Yes

5. Review Comments to the Author

Reviewer #1: Your article demonstrates a comprehensive understanding of the subject matter and presents valuable insights. The thorough research and clear presentation make it a strong contribution to the academic community.

Reviewer #2: The goal of this study was to review approaches for intensity scaling of T1w and T2w images. The authors focused on an interesting subcategory of methods that either utilize extracerebral reference regions or take T1w/T2w ratios. Based on the manuscript, the rationale of this review is 1) to examine the efficacy of these reviewed method compared to other scaling methods, and 2) to explore potential biophysical information contained in scaled T1w and T2w images.

This study has a number of strengths, including the review methodology and the thorough presentation of data. The manuscript itself is also very well written. Nonetheless, I believe the paper could benefit from further discussion of the implications of their findings, and there are a few areas in which more clarification is needed. These concerns are detailed below.

1. The inclusion of T1w/T2w ratio in this review is interesting, but it’s apparently a different approach compared to intensity scaling based on reference regions or histogram matching. I therefore think a dedicated paragraph is needed to justify this rationale and introduce commonly used terms in the literature. In particular, intensity normalization vs. intensity scaling: some people use them interchangeably, but others may have very different definitions (my impression is that scaling involves a linear transformation, such as the Ganzetti method, while intensity normalization include histogram matching and local bias correction methods).

2. Following the previous comment, taking the ratio of T1w over T2w images in my opinion serves as a bias correction (canceling out B1- field). As the authors briefly mentioned in the introduction that “An essential prerequisite is the bias field correction” and scaling is done on “conventional bias-corrected MRI”, I think more introduction is needed about bias correction and its relationship with intensity scaling in image processing. For instance, many bias correction methods exist targeting different kind of bias fields (e.g., take T1w/T2w ratio, estimate local intensity inhomogeneities, apply field maps). This will help further clarify the range of current review.

3. When referring to the Sham 2019 article, the authors present the correlation R square between R1 and “raw T1w/T2w” to be 0.8 (line 288). However, this is the results for T1w/T2w that went through additional bias correction using the group average myelin map on Conte69 mesh. The “regular” bias corrected T1w/T2w (which may be more comparable to the other T1w/T2w ratios reviewed here?) has a correlation R square of 0.74 with R1. Please clarify the definition of “raw” when describing these images.

4. A brief mention of the specificity of the quantitative measures included here would be helpful. Diffusion measures are sensitive to microstructural changes but are not specific. For myelin estimate, I think MWF should be the most robust one out of those included here. Also, I don’t think diffusion metrics and MTR are normally considered as quantitative (maybe single them out as measures related to microstructure/tissue property?).

5. In terms of the correlation with quantitative MRI or histology, I think the discussion should also point out potential biases from correlation approaches. For example, most studies used an across-region/voxel correlation (each dot in the scatter plot represents one region/voxel from one subject, number of datapoint = N(subject) * N(region)). However, some reviewed here, such as Hannoun 2022, did the correlation across subject in each region separately (number of datapoint = N(subject)). Including more datapoint increases statistical power and may have smaller r coefficient to come up as significant. Moreover, when the same subject contributes to more than one datapoint in the correlation, one may also argue that the data included are not independent, therefore violating the assumptions of Pearson’s correlation.

6. Another question not explicitly answered is whether using extracerebral reference regions really differs from or outperforms cerebral ref regions. Both may artificially affect meaningful between-subject variation, and the signal intensity can change with biological factors and pathology (this study shows T1w signal of the eyeball is also correlated with age https://www.ncbi.nlm.nih.gov/pubmed/33270843). Then do the findings of this review support the use of extracerebral ref regions over cerebral ones?

7. Lastly, the discussion would also benefit from an evaluation of approaches for checking the quality of a given scaling method. Is variance reduction the ultimate criterion of a good scaling? Or strong correlation with histology/quantitative MRI, preservation of patient vs. control differences, or elimination of site/batch effect?

Minor comments:

1. The phrasing in the abstract and introduction can be misleading. The authors state that they “summarize approaches for intensity scaling of conventional brain MRI that avoid reference regions within the brain”, which gave me the impression that this study only reviews methods that don’t involve reference regions. Conventional brain MRI is also non-specific: FLAIR and proton density are typically acquired for clinical scans but not examined here. The authors could clarify this by using more specific descriptions, like: “summarize approaches for intensity scaling of T1w and T2w images that utilize extracerebral reference regions”.

2. Table 4. Gilmore (2007) study on scaled T1w correlation with MD: better have the number right aligned to be consistent with other positive correlations.

3. Table 4. Gilmore (2007) on scaled T2w correlation with FA and MD: while the r coefficients are not available, it may be useful to indicate the direction of these correlations.

4. Line 398: T1-weighted -> T1w (abbreviation introduced previously)

6. PLOS authors have the option to publish the peer review history of their article (what does this mean?). If published, this will include your full peer review and any attached files.

Reviewer #1: **Yes: **Mahdi Mohammadi

Reviewer #2: No

---

## [Author Response · Author response to Decision Letter 0]

22 Jan 2024

Dear Dr. Udding

Dear Reviewers:

On behalf of all authors, we thank you for your positive comments. We feel we can address the critique (only raised by Reviewer 2) adequately. We hope very much that you find our new version improved so that it can be further considered for publication. We resubmitted our revised manuscript in two versions, one without and one with the relevant changes marked. Please note that line numbers in the rebuttal letter refer to the clean version.

Please find below our point-by-point response.

Kind regards

Tun Wiltgen and Mark Mühlau

Point-by-point response:

Major concerns:

‘1. The inclusion of T1w/T2w ratio in this review is interesting, but it’s apparently a different approach compared to intensity scaling based on reference regions or histogram matching. I therefore think a dedicated paragraph is needed to justify this rationale and introduce commonly used terms in the literature. In particular, intensity normalization vs. intensity scaling: some people use them interchangeably, but others may have very different definitions (my impression is that scaling involves a linear transformation, such as the Ganzetti method, while intensity normalization include histogram matching and local bias correction methods).

2. Following the previous comment, taking the ratio of T1w over T2w images in my opinion serves as a bias correction (canceling out B1- field). As the authors briefly mentioned in the introduction that “An essential prerequisite is the bias field correction” and scaling is done on “conventional bias-corrected MRI”, I think more introduction is needed about bias correction and its relationship with intensity scaling in image processing. For instance, many bias correction methods exist targeting different kind of bias fields (e.g., take T1w/T2w ratio, estimate local intensity inhomogeneities, apply field maps). This will help further clarify the range of current review.’

We have now further described the relationship of bias field correction with intensity scaling in the introduction (lines 56-60): 

“Particularly, in studies investigating image intensities, local intensity inhomogeneities induced by transmit and receiver field inhomogeneities should be removed through bias field correction because observed signal variations (after intensity scaling) should arise from biological differences of tissues rather than from technical factors.”

We agree that the T1w/T2w-ratio constitutes a different rationale and exploits a different principle than intensity scaling based on reference regions. While we had described the basic ideas of intensity scaling, we now see that the introduction needs clarification on why we included the T1w/T2w-ratio method in our review. We have now added a paragraph in the introduction dedicated explicitly to the T1w/T2w-ratio and moved parts of the last introduction paragraph to this new paragraph (lines 77-92):

“Originally, we planned to include studies using an extracerebral reference region for intensity scaling. However, the literature search revealed many studies that applied intensity scaling without reference regions but by using the T1w/T2w-ratio introduced by Glasser and van Essen [12]. We found several versions of this approach, either with or without prior scaling of the T1w and T2w images with extracerebral reference regions. Hence, we decided to extend the scope of our review and included studies using the T1w/T2w-ratio with or without prior intensity scaling if the validity of the approach was investigated. Calculation of T1w/T2w-ratio images does not require a reference region, and by dividing T1w by T2w image intensities, it is possible to obtain images with new contrasts. Most of these studies assume that the ratio of T1w and T2w images reflects myelin [12], since voxels in highly myelinated regions have high-intensity values in T1w images but low-intensity values in T2w images, but did not evaluate the method further. In addition, it has been hypothesized that T1w and T2w images are similarly affected by technical artifacts, such as bias field, and that, therefore, by calculating the ratio of T1w and T2w images, these technical artifacts are canceled out at the same time [12], [13], [14]. Yet, this assumption does not seem to be true for all MRI scanners; in consequence, some authors suggested separate scaling of T1w and T2w images before calculating the ratio of the two [14].”

During the process of this systematic review, we also found that the interchangeable use of the terms scaling and normalization can be very confusing. In our experience, intensity normalization is often used for z-score normalization (subtracting the mean and dividing by the standard deviation) or, as the reviewer mentioned, for histogram matching. In this systematic review, however, we focus on intensity scaling (i.e., translating intensity values by a determined algebraic rule, usually a linear transformation). To prevent confusion for the reader, we avoid the term intensity normalization in the text completely. It is only given in the keywords (as some authors use intensity normalization and intensity scaling interchangeably).

‘3. When referring to the Sham 2019 article, the authors present the correlation R square between R1 and “raw T1w/T2w” to be 0.8 (line 288). However, this is the results for T1w/T2w that went through additional bias correction using the group average myelin map on Conte69 mesh. The “regular” bias corrected T1w/T2w (which may be more comparable to the other T1w/T2w ratios reviewed here?) has a correlation R square of 0.74 with R1. Please clarify the definition of “raw” when describing these images.’

We agree that we should provide the data on the T1w/T2w-ratio without additional bias correction. We have now changed the value from 0.8 to 0.74. 

Table 3 in S2 Supplementary Material indicates whether and which bias field correction has been applied to the images for each study. 

In addition, we have now changed the terminology (from “raw” to “unscaled” but the changes are not marked in revised manuscript) for clarity as we think intuitively “raw” should refer to unprocessed images (as output by the scanner to the picture archiving system). We now simply refer to unscaled (but possibly bias-corrected) images as “unscaled” images, which we have now specified in section 3.2 (lines 164-167):

“We refer to “unscaled” ratios whenever image intensities have not been scaled. However, images used to calculate the ratio may have undergone bias field correction as in most studies using T1w/T2w-ratios (Table 3 in S2 Supplementary Material).”

We have also changed the terminology from “raw” to “unscaled” in the supplementary material.

‘4. A brief mention of the specificity of the quantitative measures included here would be helpful. Diffusion measures are sensitive to microstructural changes but are not specific. For myelin estimate, I think MWF should be the most robust one out of those included here. Also, I don’t think diffusion metrics and MTR are normally considered as quantitative (maybe single them out as measures related to microstructure/tissue property?).’

We added information on the diffusion measures (lines 291-293):

“The correlations with DWI metrics, known to be sensitive to microstructural changes, were statistically insignificant, except for the correlation between mean diffusivity (MD) and scaled T1w …”.

We agree that MWF should be the most robust myelin estimate. We had emphasized the relation of MWF to myelin in line 289-290 and in lines 410-412.

In section 2, we have now specified that we also consider MRI measures related to tissue microstructure (such as MTR), and we have now renamed the section, table, and figure titles so that they include: “MRI measures related to tissue microstructure”. For shortness of wording, we still refer only to quantitative MRI but we have now added the following in section 3.3.2 (lines 241-243):

“Although we are aware that they are not fully quantitative, we refer to the “MRI measures related to tissue microstructure”, such as magnetization transfer ratio (MTR) or diffusion-weighted imaging (DWI), as “quantitative imaging” for readability.”

‘5. In terms of the correlation with quantitative MRI or histology, I think the discussion should also point out potential biases from correlation approaches. For example, most studies used an across-region/voxel correlation (each dot in the scatter plot represents one region/voxel from one subject, number of datapoint = N(subject) * N(region)). However, some reviewed here, such as Hannoun 2022, did the correlation across subject in each region separately (number of datapoint = N(subject)). Including more datapoint increases statistical power and may have smaller r coefficient to come up as significant. Moreover, when the same subject contributes to more than one datapoint in the correlation, one may also argue that the data included are not independent, therefore violating the assumptions of Pearson’s correlation.’

We fully agree. Yet, this problem does not directly influence our findings as we could only review and summarize data descriptively. We now mention this issue in the discussion (lines 451-458): 

“We tried to collect and present results as homogeneously as possible. However, some methodological differences regarding correlation analyses exist. Some studies calculated correlations with quantitative MRI or histology across subjects and regions, and some calculated the correlations across subjects in each region individually. This seemingly little difference may have influenced the results, however. First, including more regions, and, hence, more data points, may lead to increased statistical power; second, if the same subject contributes to more than one data point, the assumptions of Pearson’s correlation are potentially violated.”

‘6. Another question not explicitly answered is whether using extracerebral reference regions really differs from or outperforms cerebral ref regions. Both may artificially affect meaningful between-subject variation, and the signal intensity can change with biological factors and pathology (this study shows T1w signal of the eyeball is also correlated with age https://www.ncbi.nlm.nih.gov/pubmed/33270843). Then do the findings of this review support the use of extracerebral ref regions over cerebral ones?’

We had addressed this in the conclusion. Again, we must emphasize that we do NOT believe that scaling with extracerebral reference regions is always the better choice. A huge caveat is that even the extracerebral region can be affected by neuropsychiatric diseases. In the worst case, this association may be unknown! In the discussion (lines 473-475), we had addressed this issue with the example of the temporal muscle; we have now added the reviewer’s impressive example of the eyeball by Streckenbach et al. (PMID 33270843), which we had not known. 

‘7. Lastly, the discussion would also benefit from an evaluation of approaches for checking the quality of a given scaling method. Is variance reduction the ultimate criterion of a good scaling? Or strong correlation with histology/quantitative MRI, preservation of patient vs. control differences, or elimination of site/batch effect?’

We fully agree. In the discussion, we mentioned variance reduction as a criterion for intensity scaling, and we concluded that maximizing variance reduction can be misleading since biologically induced variance might also be removed. Elimination of site/batch effect can also be a reasonable goal, but good results in this regard do not necessarily imply good intensity scaling since it might also lead to the removal of biologically induced variance. We now elaborate more clearly that we rate variance reduction as a necessary but not a sufficient condition for valid intensity scaling (lines 390-391):

“Hence, variance reduction is a necessary but not a sufficient condition for evaluating intensity scaling methods.” 

Moreover, we also see an important role in correlation with histology, quantitative MRI (and MRI-measures related to microstructure), and preservation of associations with clinical parameters (e.g., group differences and correlations with clinical scores). The same applies to the elimination of site/batch effects, which was not investigated in the studies we identified, however. We now mention these opportunities in the discussion (lines 477-482): 

“Scaling methods should be evaluated according to available evidence. We regard correlation with histology or quantitative MRI (or MRI measures related to tissue microstructure) as most desirable, followed by preservation or demonstration of known associations with clinical parameters such as group differences or correlations with clinical scores. In contrast, evidence based on mere variance reduction is precarious as it can go along with removing biologically important variance.”

Minor concerns:

‘1. The phrasing in the abstract and introduction can be misleading. The authors state that they “summarize approaches for intensity scaling of conventional brain MRI that avoid reference regions within the brain”, which gave me the impression that this study only reviews methods that don’t involve reference regions. Conventional brain MRI is also non-specific: FLAIR and proton density are typically acquired for clinical scans but not examined here. The authors could clarify this by using more specific descriptions, like: “summarize approaches for intensity scaling of T1w and T2w images that utilize extracerebral reference regions”.’

We, in part, agree. We now use the term T1w and T2w images instead of conventional MRI. Regarding the term ‘extracerebral reference region,’ we see the problem that it does not really cover T1w/T2w-ratio images. Therefore, we now write ‘intensity scaling of conventional T1-weighted (w) and T2w brain MRI avoiding reference regions within the brain’ (line 21).

‘2. Table 4. Gilmore (2007) study on scaled T1w correlation with MD: better have the number right aligned to be consistent with other positive correlations.’

The correlation of T1w with MD is negative, which is why we had aligned the number on the left side. The trend of the correlation can be deduced from Fig 5 in the supplementary material of Gilomore’s publication.

‘3. Table 4. Gilmore (2007) on scaled T2w correlation with FA and MD: while the r coefficients are not available, it may be useful to indicate the direction of these correlations.’

We have now added “positive” or “negative” in the “Results” column.

‘4. Line 398: T1-weighted -> T1w (abbreviation introduced previously)’

We have now changed T1-weighted to T1w.

---

## [Decision Letter · Decision Letter 1]

29 Jan 2024

Intensity scaling of conventional brain magnetic resonance images avoiding cerebral reference regions: A systematic review

PONE-D-23-37143R1

Dear Dr. Wiltgen,

We’re pleased to inform you that your manuscript has been judged scientifically suitable for publication and will be formally accepted for publication once it meets all outstanding technical requirements. 

Authors are encouraged to address the minor comments provided by reviewer 2. 

Kind regards,

Md Nasir Uddin, PhD

Academic Editor

PLOS ONE

Additional Editor Comments (optional):

Reviewers' comments:

Reviewer's Responses to Questions

**Comments to the Author**

1. If the authors have adequately addressed your comments raised in a previous round of review and you feel that this manuscript is now acceptable for publication, you may indicate that here to bypass the “Comments to the Author” section, enter your conflict of interest statement in the “Confidential to Editor” section, and submit your "Accept" recommendation.

Reviewer #1: All comments have been addressed

Reviewer #2: All comments have been addressed

2. Is the manuscript technically sound, and do the data support the conclusions?

Reviewer #1: Yes

Reviewer #2: Yes

3. Has the statistical analysis been performed appropriately and rigorously? 

Reviewer #1: Yes

Reviewer #2: Yes

4. Have the authors made all data underlying the findings in their manuscript fully available?

Reviewer #1: Yes

Reviewer #2: Yes

5. Is the manuscript presented in an intelligible fashion and written in standard English?

Reviewer #1: Yes

Reviewer #2: Yes

6. Review Comments to the Author

Reviewer #1: (No Response)

Reviewer #2: The authors have suitably addressed my previous comments. Based on their changes, I have only one minor comment that I think might benefit the manuscript:

Revised line 87-92: please make clear that by taking T1w/T2w ratio, only the receiver bias field (B1-) can be exactly canceled out, given that the head position is not changed. B1+ fields are different in T1w and T2w images due to pulse differences and these inhomogeneities in extreme situations (like in 7T) may even affect image contrast (e.g., in a single image, GM/WM contrast is higher in the center and lower in peripheral areas of FOV. This is also why in my opinion integrating data acquired with different protocol is probably never feasible/accurate no matter which scaling method is used, but this may be outside the scope of current discussion).

Thank you to the authors for looking into this interesting topic and I look forward to citing this in the future.

7. PLOS authors have the option to publish the peer review history of their article (what does this mean?). If published, this will include your full peer review and any attached files.

Reviewer #1: **Yes: **Mahdi Mohammadi

Reviewer #2: No

---

## [Editor Report · Acceptance letter]

12 Feb 2024

PONE-D-23-37143R1 

PLOS ONE

Dear Dr. Wiltgen, 

I'm pleased to inform you that your manuscript has been deemed suitable for publication in PLOS ONE. Congratulations! Your manuscript is now being handed over to our production team.

Kind regards, 

on behalf of

Dr. Md Nasir Uddin 

Academic Editor

PLOS ONE